# Racial Differences in Breastfeeding on the Mississippi Gulf Coast: Making Sense of a Promotion-Prevalence Paradox with Cross-Sectional Data

**DOI:** 10.3390/healthcare10122444

**Published:** 2022-12-03

**Authors:** John P. Bartkowski, Janelle Kohler, Xiaohe Xu, Tennille Collins, Jacinda B. Roach, Caroline Newkirk, Katherine Klee

**Affiliations:** 1Department of Sociology, University of Texas at San Antonio, San Antonio, TX 78249, USA; 2Department of Psychology, University of Texas at San Antonio, San Antonio, TX 78249, USA; 3Mississippi Department of Human Services, Jackson, MS 39201, USA; 4Mississippi Public Health Institute, Ridgeland, MS 39157, USA

**Keywords:** breastfeeding, lactation, race, ethnicity, African American, Mississippi, social climate, rural, health promotion, peer support, disparities

## Abstract

Breastfeeding is less prevalent among African American women than their white peers. Moreover, breastfeeding rates in the South lag behind those in other regions of the U.S. Consequently, various efforts have been undertaken to promote breastfeeding among groups for which this practice is less common. This study examines African American and white racial disparities concerning (1) exposure to breastfeeding promotional information and (2) reported prevalence of breastfeeding in primary social networks. The survey combines a randomly selected sample of adults representative of the population and a non-random oversample of African Americans in a predominantly rural tri-county area on the Mississippi Gulf Coast. An initial wave of 2019 Mississippi REACH Social Climate Survey data collected under the auspices of the CDC-funded REACH program (Mississippi’s Healthy Families, Mothers, and Babies Initiative; 2018–2023) is used to examine racial disparities in these two key outcomes for Mississippians in Hancock, Harrison, and Jackson counties. The results show that African American respondents are more likely to be exposed to breastfeeding promotional messages than their white counterparts. However, the reported prevalence of breastfeeding in African American respondents’ primary social networks is significantly lower than that indicated by their white peers. These paradoxical results underscore the limitations of promotional efforts alone to foster breastfeeding. While breastfeeding promotion is important, the reduction of racial disparities in this practice likely requires a multi-pronged effort that involves structural breastfeeding supports (e.g., lactation spaces, peer networking groups, and pro-breastfeeding employment policies and workplaces). This study provides a promising model of innovative methodological approaches to the study of breastfeeding while underscoring the complex nature of racial disparities in lactation prevalence.

## 1. Introduction

The American Academy of Pediatrics recommends that infants be exclusively breastfed for the first 6 months of life and continue to breastfeed with complementary foods until 12+ months [1]. Healthy People 2030 breastfeeding objectives include increasing the proportion of infants who are (1) breastfed at 1 year, and (2) breastfed exclusively through 6 months, along with the expansion of worksite lactation programs [2,3]. Despite strong recommendations to breastfeed, rates of African American women’s breastfeeding initiation and duration are the lowest in the U.S. compared with those of other racial-ethnic groups [4,5]. African Americans represent 15.2% of Women, Infant, and Children (WIC) breastfeeding women [6]. Rates in breastfeeding differ by U.S. state, with lower rates among African Americans in most states. There are some isolated cases of exceptionally high African American breastfeeding rates in the northeast (Massachusetts, Rhode Island) and elsewhere (Minnesota, Georgia, Arizona) [1]. Nevertheless, for those who exclusively breastfed through 6 months, there are significantly lower rates among African American infants in 22 states. In 12 states, there was a difference of at least 10 percentage points in exclusive breastfeeding between African Americans and whites through 6 months; similar differences were observed in 22 states at 12 months [1]. These racial disparities are further amplified by regional differences, such that women in Southern states tend to breastfeed at lower rates than their peers elsewhere [4,7]. Meyerink and colleagues examined 150 mothers in the Southeastern U.S., 93% of which were African American, and found that only 41% of study participants had initiated breastfeeding, 24% breastfed for at least 1 month, and 8.3% had breastfed for 3 or more months [8]. In that same investigation, a positive relationship with breastfeeding was observed when the mother herself had been breastfed while a negative relationship between breastfeeding and having a premature infant was evident. Further predictors of breastfeeding among women in this study were age, social network influences, and prior experiences. Specifically, breastfeeding at 1 month was more common among women who were older and knew close relatives who breastfed. Those who had previously breastfed were also more inclined to do so again. Notably, income and education were found to buffer barriers to breastfeeding. Of all states investigated, Mississippi had the lowest rates of breastfeeding at 6 months and 1 year [8,9,10]. This attribute, along with the state’s generally pronounced racialized health disparities, make Mississippi an excellent site at which to discern the contributors to racial differences in breastfeeding prevalence.

Despite some positive predictors of breastfeeding initiation and duration among African American women [4,7], obstacles include interpersonal (e.g., psychosocial, family, peer network), sociocultural (historical, ethnic), and institutional (e.g., economic, political) factors [11,12,13,14]. Racism, bias, and discrimination have also been identified as formidable breastfeeding barriers [5] and may be coupled with jobs that lack maternity leave and do not accommodate breastfeeding [10,15,16]. For example, African American women more often report a lack of maternity leave from work when compared with their peers in other racial-ethnic groups [11]. Typically, women in the U.S. return to work between 3 and 6 months postpartum [17]. However, African American women generally return to work after 8 weeks and are more likely to hold jobs that do not welcome breastfeeding [16]. Some evidence suggests that returning to the workplace is a primary barrier for African American women to breastfeed successfully [7]. Additional breastfeeding barriers faced by African Americans include a lack of access to electric pumps, social pressures to supplement with formula, and a lack of breastfeeding role models and support networks [10], as well as food insecurity and infant trouble sucking or latching [18]. In a qualitative study conducted by Shipp and colleagues, 13 African American women were interviewed after the conclusion of a postpartum support program that followed breastfeeding initiation [19]. Participants shared common challenges to breastfeeding which included uncertainty about whether their infant was receiving enough nutrition, latching and pumping difficulties, and an overall lack of comfort with breastfeeding in public due to family and others. The results indicated that breastfeeding education along with peer and health professional support can be important influences on exclusive breastfeeding.

Healthcare providers sometimes assume that African American women will not breastfeed, and this presumption can undermine breastfeeding support (e.g., lactation support referrals, problem-solving assistance, WIC counselors, lactation providers) [4,20]. Focus group data have revealed that African Americans often receive inadequate breastfeeding support from key points of contact, including WIC counselors, physicians, and lactation providers [20]. Some WIC counselors are said to be quick to recommend and provide formula, even as physicians were reported to ignore African American women’s infant feeding priorities. These same focus group interviewees charged that lactation support providers often lacked vital knowledge and information, were critical of how a mother was trying to breastfeed, and seemed rushed or unavailable while women were at the hospital.

Broader social-ecological factors also loom large [14]. There is a troubling interplay of social factors at the interpersonal level (i.e., women who live with extended family are less likely to breastfeed), institutional level (i.e., women who return to blue-collar jobs are less inclined to breastfeed), and sociocultural level (i.e., longstanding cultural beliefs about human milk dating back more than a century). These combined forces contribute to negative perceptions of breastfeeding. Interestingly, within the macrosystem, Reeves and Woods-Giscombé explained how African American women felt uncomfortable breastfeeding because nowadays, “women’s bodies and breasts are ubiquitously used to sell food, alcohol, automobiles, films, and a myriad of other products and services” [14]. The racialized sexualization of women’s breasts can therefore undermine breastfeeding [4].

Some research has explored avenues for rectifying racial disparities in breastfeeding. Studies have revealed that increasing the time allotment spent with patients, engaging in follow-up care, while also expanding the availability of staff training and attention are effective in reducing racial health disparities and enhancing the initiation and duration of breastfeeding [6,21]. Given very low breastfeeding rates in Mississippi and surrounding states, some interventions stressing skin-to-skin care and rooming-in have proven remarkably effective [7,9]. Merewood and colleagues reported positive results from Ten Steps to Successful Breastfeeding, which was implemented in Mississippi, Louisiana, Tennessee, and Texas [4,22]. Specifically, the disparity in breastfeeding initiation between African Americans and whites decreased by 9.6 percentage points over 31 months. For all races combined, breastfeeding initiation increased from 66% to 75% while the proportion of those exclusively breastfeeding rose from 34% to 39%. For African Americans, breastfeeding initiation increased significantly (46% to 63%, *p* < 0.05), as did exclusive breastfeeding (19% to 31%, *p* < 0.05). Furthermore, there was a significant association for all races between skin-to-skin care after cesarean delivery and breastfeeding initiation and exclusivity, as well as a significant association between rooming-in and exclusive breastfeeding among African Americans [4,22]. Another facilitator for breastfeeding among African American women was support groups. Support groups have yielded increased knowledge of breastfeeding among participants along with social reinforcement through peer networks [23].

The WHO has an International Code of Marketing Breast-milk Substitutes and offers guidelines aimed at ending what they call the “inappropriate promotion of food for infants and young children” [24]. A systematic review of prior research explored the impact of social media on breastfeeding practices [25]. While mixed results emerged, it was apparent that social media can be enlisted to improve awareness and attitudes about breastfeeding. Some research has assessed how various media sources (i.e., internet, television, radio, newspapers, and outdoor posters/billboards) may promote breastfeeding [26,27]. Persons reporting message exposure from a medium number of media sources had higher odds ratios for initiation, any breastfeeding at 2 months, and exclusive breastfeeding at 2 months. However, such research has not reported on the differences in types of media exposure, nor did it examine differences between exposure to media messages and race.

One intervention that was conducted across three counties in rural Mississippi was Delta Healthy Sprouts [28]. The majority of participants were African American women (79 out of 82; 96.3%). This intervention produced mixed results. While there was a significant increase in breastfeeding knowledge scores from baseline to the late gestational period, breastfeeding belief scores were higher for those who initiated breastfeeding compared to those who did not. Overall, this intervention revealed the inadequacy of increasing knowledge and addressing modifiable barriers (i.e., social norms, lack of family and social support, embarrassment, lactation issues, and employment and childcare) without additional and sustainable support modifications. Thomson and colleagues concluded that “improving breastfeeding outcomes for all socioeconomic groups will require consistent, engaging, culturally relevant education that positively influences beliefs as well as social and environmental supports that make breastfeeding the more accepted, convenient, and economical choice for infant feeding” [28]. Therefore, a holistic set of promotional efforts and supports are needed to maximize the impact of pro-breastfeeding interventions.

No research of which we are aware has examined the relationship between African Americans and the promotion of breastfeeding from specific media sources. Our study differs from Delta Healthy Sprouts in that we are assessing multiple pro-breastfeeding media sources which African Americans may have encountered compared to their white peers at not only a broad level (i.e., television), but also in the nearby area (i.e., local pharmacy). Specifically, our study examines African American-white racial disparities concerning (1) exposure to breastfeeding promotional information and (2) reported prevalence of breastfeeding in primary social networks.

## 2. Materials and Methods

### 2.1. Research Design

A cross-sectional survey design was used to predict exposure to breastfeeding promotional information and reported prevalence of breastfeeding in primary social networks. This design was chosen because the Mississippi REACH Social Climate Survey is a baseline assessment of an overall project that continues through 2023 (see Appendix A link for survey report). This snapshot was intended to elucidate the contours and antecedents of African American-white disparities in breastfeeding along the Mississippi Gulf Coast. Data collection approval was received and governed by an existing memorandum of understanding between the Declaration of Helsinki and the Institutional Review Board of Mississippi State University (IRB-17-04-MOU).

### 2.2. Setting and Relevant Context

Participants completed the Mississippi REACH Social Climate Survey from the Mississippi Public Health Institute [29,30]. The Mississippi REACH Social Climate Survey was funded by the Centers for Disease Control and Prevention’s Racial and Ethnic Approaches to Community Health (CDC REACH) program, which supported the implementation of the Healthy Families, Mothers, and Babies Initiative [31,32,33]. The Mississippi REACH Social Climate Survey examined racial disparities for residents in Hancock, Harrison, and Jackson counties along the Mississippi Gulf Coast, where poverty and other adverse social determinants of health are prevalent, thereby elevating the mortality and morbidity risks in this area. Such health risks are further magnified for African American residents. African American representation in these counties varies, respectively, at 8.7% (Hancock), 26.3% (Harrison), and 21.6% (Jackson). County poverty rates are relatively high in Hancock (15.6%), Harrison (16.8%), and Jackson (13.4%) counties, and the African American poverty rate in Mississippi is about 2.5 times that of whites [34].

### 2.3. Data Collection and Sample

In collaboration with the Mississippi State University Social Science Research Center, researchers from the Mississippi Public Health Institute designed a telephone-based survey to measure the behaviors and opinions of African Americans and Mississippians in the Gulf Coast region regarding nutrition, tobacco use, and infant feeding. A representative sample of adults aged 18 and older was drawn from three counties (Hancock, Harrison, and Jackson) in the state of Mississippi from July through September 2019 (Table 1). The sample was drawn from the cellular telephone universe with an additional oversample filter of African Americans aged 18–50. To ensure that a reproducible and representative sample was obtained, independent probability-based samples were selected via a random digit dialing (cellphone-only sampling) methodology. In total, 20,000 general population and 60,000 oversample numbers were selected from a universe of 1,083,000. Telephone numbers were dialed for a maximum of eight times before being retired, unless a final disposition was secured prior to the eighth call.

This effort secured 307 completed surveys from the catchment area. Additional surveys were completed as part of an oversample of African Americans aged 18–50. This oversample relied on two screening items that excluded respondents from the general population who did not meet the targeted demographic criteria of age and race. The oversample secured an additional 112 completed surveys. (There was a cooperation rate of 38.7% for the general population and 7.9% for the oversampled population. The cooperation rate is defined as completes/(completes + refusals).

An inclusive sampling method was used for black males and females 18 to 50 years of age residing in the Gulf Coast region of Mississippi, including Jackson, Hancock, and Harrison counties. The first wave of the Mississippi REACH Social Climate Survey was completed by 419 participants in 2019. (The second wave of data is anticipated to be collected in 2023.) The majority of the participants were white (*n* = 224, 53.41%), followed by African American (*n* = 173, 41.29%), and then other races (*n* = 22, 5.25%). The average age of participants was 42.98 years (SD = 16.0). The majority of participants reported being female (*n* = 255, 60.86%), and employed, (*n* = 261, 62.24%). For a demographic breakdown of employment status and education, see Table 1 and Table 2.

### 2.4. Measurement

#### 2.4.1. Exposure to Breastfeeding Promotion

Exposure to breastfeeding promotional message items were sourced from the Consumer Opinion Panel, which has been used in other studies [35,36]. Each of the 11 items began with “Have you recently seen, heard, or read anything about breastfeeding from the following…” The 11 sources that followed this question stem were (1) Television, (2) Magazine, (3) Newspaper, (4) Radio, (5) Social media, (6) Online, but not social media, (7) Billboards or outdoor posters, (8) Local pharmacy, (9) Local church or faith community, (10) Doctor’s office or other healthcare center, and (11) Local civic organization such as a women’s group or family resource center. Each question had four response categories: Yes, No, Don’t know, and Refused. These responses were dummy-coded with “Yes” = 1, “No” = 0, and other responses = missing. These items were summed to form a count index variable to indicate the number of media sources. The reliability of this count index variable was acceptable (Cronbach’s alpha = 0.790).

#### 2.4.2. Prevalence of Breastfeeding in Primary Social Networks

A global indicator was used to measure breastfeeding by women within the survey respondent’s primary social networks. This measure was adapted, with slight revisions, from the Consumer Opinion Panel and was a single item that read, “How common is breastfeeding infants among your female friends and relatives?” This item has the benefit of being answerable by both women and men. Participants could select one response from among ten options, as follows: (1) None of them have breastfed, (2) Very few of them have breastfed, (3) About one quarter of them have breastfed, (4) About half of them have breastfed, (5) A majority of them have breastfed, (6) All of them have breastfed, (7) None of my female friends or relatives have any children, (8) I don’t have any female friends or relatives, (9) Don’t know, and (10) Refused. To assess the prevalence of breastfeeding within primary social networks, only the first six responses were analyzed (*n* = 380).

#### 2.4.3. Predictor and Control Variables

The main predictor variable of interest was race. Control variables included employment status, education, household income, gender, and age. Race was dummy-coded with white serving as the reference category, while African American and other races were compared. This approach is intentionally adopted to highlight African American differences, our key point of interest. Employment status was dummy-coded into a dichotomous variable with unemployed serving as the reference category. Age was treated as a continuous variable (in years). The education variable measures the last grade attended ranging from 1 = never attended school or only attended kindergarten to 8 = Completed master’s degree. As featured in the General Social Survey, income was measured in income brackets, which has been routinely used as a continuous variable in sociological research. When specified as a categorical variable, our findings were largely unaffected. For example, the regression coefficients for African American respondents remained significant in both models. All participants reported either identifying as a man or woman; hence, gender was dichotomized into male and female, with male being the reference category.

### 2.5. Data Analysis

We used the mean, standard deviation, frequency, and percentage to describe characteristics of the study sample. We then compared racial differences in terms of sociodemographic characteristics by using the chi-square statistic for categorical variables and the ANOVA F statistic for continuous variables, respectively. Bar charts were used to show the differences in the distribution of the types of promotional encounters and the count of sources of media exposure by race. Our research team first assessed the differences reported in breastfeeding media message reception among African American and white Mississippians on the Gulf Coast, after which we examined the prevalence of breastfeeding infants among respondents’ female friends and relatives. As noted, this social network breastfeeding prevalence measure has the virtue of securing answers from both men and women survey respondents. Next, two multivariate regression models were estimated to examine if race predicted both breastfeeding promotional information (a negative binomial regression model) and prevalence of breastfeeding in primary social networks (a linear regression model) net of sociodemographic controls. The breastfeeding prevalence in primary social networks variable was treated as a continuous variable. Before conducting regression analysis, the iterative Markov chain Monte Carlo (MCMC) multiple imputation method in SPSS was used to replace missing values. Regression results prior to imputation are available by request but are not currently included. The *p*-value < 0.05 was set as the threshold of statistical significance.

## 3. Results

Table 2 features the percentage of respondents for each income category included in the survey. The most frequent responses (at roughly 11% each) were found in two middle-income categories ($35,000 to under $50,000 and $50,000 to under $75,000). Figure 1 provides a visual depiction of percentages associated with a key dependent variable for our study, namely, promotional encounter percentages. The percentage of African Americans who report breastfeeding media exposure eclipses that of their white counterparts for every media source surveyed. More specifically, Figure 1 demonstrates that African Americans reported receiving greater breastfeeding promotional messaging from such critical sources as television, healthcare professionals, the internet, magazines, their local pharmacy, and local civic organization. Figure 2 provides a count of breastfeeding promotion exposure incidents. Whites are more likely to report no media exposure than their African American peers (69 whites versus 27 African Americans), and single media exposure is more than twice as common among African Americans than whites (48 versus 18, respectively). However, when assessing the differences between how common breastfeeding infants was among their female friends and relatives, whites reported a much higher percentage (46% compared to 24%) for the statement “Most or all friends and relatives have breastfed.” Additionally, pertaining to the statement, “Very few or no friends and relatives have breastfed,” African Americans exhibited a much higher percentage (48% compared to 22%). These sets of results introduce an interesting paradox: namely, despite more pro-breastfeeding messages received by African American Mississippians, they are less likely to have primary network contacts who have breastfed.

Quite notably, the negative binomial regression model that predicted exposure to breastfeeding promotional information was statistically significant with Likelihood Ratio χ^2^ = 41.627 and *p* < 0.001. Specifically, being an African American significantly (*p* < 0.01) increased the expected rate of media exposure by 31.78% ((e^0.276^−1) × 100), net of statistical controls. Additionally, gender and age were significantly associated with the rate of media exposure (Table 3).

Table 4 presents results of the linear regression model to predict the prevalence of breastfeeding in primary social networks. This model significantly predicted the prevalence of breastfeeding in primary social networks with F(7, 412) = 8.146, *p* < 0.001, and ΔR^2^ = 0.116. Specifically, African Americans reported having a significantly lower prevalence of breastfeeding in their primary social networks (β = −0.311, *p* < 0.001). Additionally, those who were employed reported having more prevalence of breastfeeding in their primary social networks (β = 0.137, *p* < 0.05), possibly due to more social engagement overall or perhaps network selectivity (employed persons). These results support the contention that African Americans are embedded in social networks where breastfeeding is less prevalent. This finding provides compelling evidence of lower breastfeeding prevalence among African American women on the Mississippi Gulf Coast, albeit as ascertained through the accounts of women and men survey respondents.

## 4. Discussion

Our study highlights an interesting paradox. While African Americans on the Mississippi Gulf Coast indicated greater exposure to breastfeeding promotional messages, they also reported a lower prevalence of breastfeeding in their primary social networks compared to their white counterparts. What might account for this paradox? One possibility is that breastfeeding promotion occurs within a larger media landscape that includes formula marketing. Previous studies have indicated that formula marketing even as a supplement, a factor for which we cannot account, can limit WIC’s ability to promote breastfeeding [11,37].

A second and not mutually exclusive consideration concerns structural barriers to breastfeeding faced by many African American women. African American women have limited access to electric pumps, which are crucial for sustained breastfeeding among women in the paid workforce but also often lack maternity leave and are employed in workplaces that are typically not welcoming of breastfeeding [11,15,16]. Additional obstructions may arise through lower levels of maternal education (mothers’ years of schooling), lack of breastfeeding education, and social environmental influences outside of professional development [38,39]. Despite efforts to change attitudes, practical circumstances matter. These structural factors can create a cycle in which breastfeeding is problematic and few role models or supportive social networks are available [11].

A third potential barrier to African American breastfeeding would be maternal age and physical health. Physical health and breastfeeding issues have a predominant predictor role, such as the presence of pain when feeding, as well as fluctuation, decrease, or lack of milk supply, or latching issues, all of which can be exacerbated by maternal age but countered by breastfeeding education [38,39,40]. Additionally, overweight women of various racial and ethnic backgrounds are less likely to initiate breastfeeding and more likely to breastfeed for a shorter duration than their normal-weight peers [41].

Our study builds on previous research to underscore the social determinants of health and the complex nature of breastfeeding with particular attention to this health disparity among African Americans. Using the social-ecological approach, prior scholarship has aimed to explain African American women’s lower rates of breastfeeding [14]. The social-ecological approach considers how a range of social factors within one’s microsystem, exosystem, and macrosystem can affect breastfeeding practices. The model highlights four social-ecological pathways through which decisions are made: personal, socioeconomic, psychosocial, and cultural. This model’s results bridge a gap not thoroughly addressed in prior scholarship, which lends support to the observation that African Americans do report more experience with breastfeeding promotional material. However, without a pro-breastfeeding presence offered by health professionals or social networks, promotional material does not reach its full potential. Media messaging aims to foster breastfeeding through the cultural pathway, but it is possible that an interplay of forces in other pathways combines to supersede the effects of pro-breastfeeding media reception. For example, negative personal experiences (e.g., infant difficulty latching) or socioeconomic circumstances (e.g., financial strain, no maternity leave) may be more influential than cultural messages that encourage breastfeeding. In this sense, culture is only one facet of a broader social landscape. Moreover, within the cultural pathway, media channels can send mixed messages. Consequently, public service messages that promote breastfeeding may be counteracted by exposure to infant formula advertising. In addition, historical factors should not be ignored, given that breastfeeding prevalence may be subject to long-term social disadvantages that include the legacy of racism in education and employment along with disjointed interventions [7]. In short, the effects of cultural change are likely to be blunted if structural impediments to breastfeeding remain in place.

### 4.1. Implications for Policy and Practice

The need for increased community and health systems support of breastfeeding for African American women is clarified through the findings of this study. Future directions for health policy implementation and community practice include tailored breastfeeding education and recognizing the circumstances that foster African American women’s breastfeeding (e.g., workplace supports). The focus should surpass simple messages concerning the benefits of breastfeeding and breastfeeding initiation encouragement, with an emphasis on the technical practices and issues that may arise during breastfeeding. These challenges may include how and when to use electrical pumps, as well as the fluctuation of milk supply between feedings and the uneven supply between breasts, which can be affected by the use of pumps; it is also vital to know whom to contact with confidence that someone is within reach should questions or concerns develop [19]. WIC program refinement (i.e., more frequent staff trainings using culturally appropriate educational materials, consistent support and resources offered across locations, readily available hands-on assistance not limited to technical information) would further benefit breastfeeding initiation and duration, as women’s experience with WIC varies based on location. Additional promising directions include the encouragement and availability of breastfeeding support networks within African American communities. Social networks (i.e., family, friends, coworkers) are often an influential factor for breastfeeding initiation and duration among African American mothers. The inclusion of lactation spaces and other breastfeeding-friendly environmental changes to professional and public spaces would resolve several longstanding barriers to breastfeeding. Addressing the benefits of breastfeeding, alongside the provision of tangible resources for questions, could further aid in normalizing natural lactation and increase the amount and frequency of African American women breastfeeding.

### 4.2. Limitations and Future Directions

Like all investigations, this study has limitations, each of which can be addressed through additional research. First, this study is a baseline assessment. As such, our investigation provides an important snapshot of racial disparities in breastfeeding among whites and African Americans along the Mississippi Gulf Coast. However, our study relies on cross-sectional data. While we have identified variable associations, we cannot speak to trends or causal patterns. Mississippi REACH will continue through 2023, and a follow-up (repeated cross-sectional) wave of the Mississippi REACH Social Climate Survey is planned before the grant period ends. While important insights have emerged from this baseline assessment (focused on attitudes prior to intensive breastfeeding promotion and support initiatives), there are additional prospects to complement this portrait with follow-up data. Future assessments can help the Mississippi Public Health Institute and CDC REACH understand possible changes in breastfeeding on the Mississippi Gulf Coast.

Second, this study reported on quantitative data with an emphasis on structural barriers to breastfeeding. The field of breastfeeding support could certainly benefit from the application of qualitative methodologies. Individual and group interviews related to breastfeeding initiation, duration, and support received by African American women, as well as lactation and latching issues experienced could deepen understandings of racial differences in breastfeeding [19,38]. Further, physical breastfeeding issues experienced by African American women could enrich support services by helping to shape solutions-focused interventions designed to overcome these specific barriers. Additionally, textual analyses of promotional messages combined with assessments of message decoding processes by their intended audiences could illuminate noteworthy facets of preference and value formation.

Third, this survey was conducted prior to the COVID-19 pandemic. Data reported here may have changed significantly due to the pandemic. Widespread unemployment may foster breastfeeding prevalence due to an increased need for social services (i.e., WIC), more receptivity to breastfeeding promotional messages, and the removal of workplace breastfeeding constraints. Then again, breastfeeding may have decreased with rising food insecurity and increased widespread socioeconomic dislocation related to the COVID-19 pandemic. Empirical investigations of the pandemic’s direct and racialized effects on birth outcomes and perinatal care inequities [42], especially in relation to overall health and mortality, are also needed given COVID-magnified African American health vulnerabilities [43]. Future research should consider the pandemic’s impact on breastfeeding and the broad spectrum of messages about breastfeeding and formula. The role of structural impediments (e.g., no maternity leave) and structural facilitators (e.g., lactation support services, breastfeeding-friendly hospitals) should also be more carefully explored. Finally, our sample size does not permit us to explore the intersectionality between race and class (e.g., household income), but such topics should be considered going forward. While these questions cannot be answered through our investigation, our study has demonstrated that media messaging alone is not the key to bolster breastfeeding rates among African American women on the Mississippi Gulf Coast.

## 5. Conclusions

This study examined racial disparities among African American and white Mississippians living on the Gulf Coast concerning (1) exposure to breastfeeding promotional information and (2) reported prevalence of breastfeeding in primary social networks. The results highlight an intriguing paradox. While African Americans tend to receive more breastfeeding promotional messages, the reported prevalence of breastfeeding in their primary social networks is considerably lower than that indicated by their white counterparts. These results underscore the limitations of promotional efforts alone to foster breastfeeding. While breastfeeding promotion is important, efforts to reduce racial disparities in this practice should adopt a multi-pronged approach that involves structural breastfeeding supports (e.g., lactation spaces, peer networking groups, and pro-breastfeeding employment policies). Aside from these empirical findings, this study highlights the value of novel methodological approaches, which include an African American oversample and a network prevalence of breastfeeding indicator suitable for women and men survey respondents. Future research could aim to replicate these results among populations that exhibit low breastfeeding prevalence outside of the Mississippi Gulf Coast.

## Figures and Tables

**Figure 1 healthcare-10-02444-f001:**
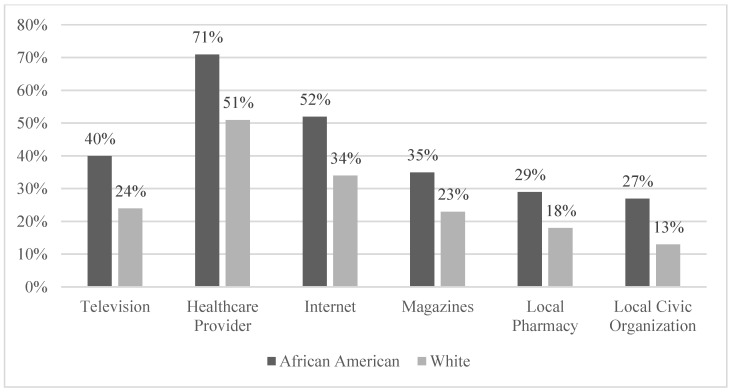
Breastfeeding media exposure.

**Figure 2 healthcare-10-02444-f002:**
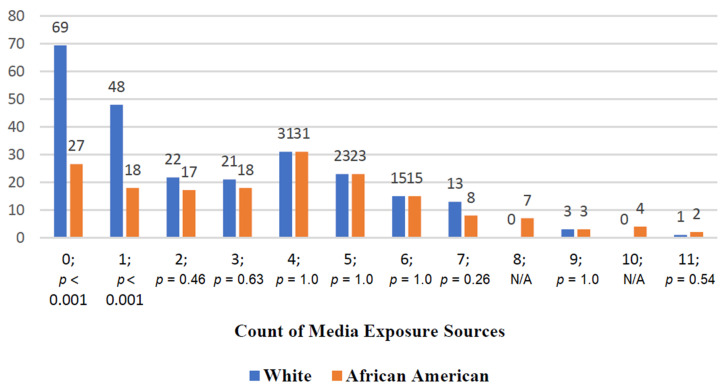
Count of media exposure sources by race. The numbers “0–11” refer to the number of sources through which media exposure to breastfeeding promotional messages was received. The majority of the variability is found in the “0” and the “1” media source categories (*p* < 0.001). These results are the basis for the negative binomial regression model.

**Table 1 healthcare-10-02444-t001:** Sample characteristics by race (*n* = 419).

	White		African American		Other Race		
	Mean	SD	Mean	SD	Mean	SD	Statistical Significance (*p*-Value)
Exposure to Breastfeeding Promotion (index)	2.5	2.4	3.7	2.7	2.8	2.6	<0.001
Prevalence of Breastfeeding in Primary Social Networks	3.9	1.5	3.0	1.6	4.0	1.7	<0.001
Race ^a^	-	-	-	-	-	-	-
White (reference)	223.8	53.4	-	-	-	-	-
African American	-	-	173.0	41.3	-	-	-
Other Races	-	-	-	-	22.0	5.3	-
Employment Status ^a^	-	-	-	-			0.025
Yes	127.6	57.0	121.0	69.9	12.2	55.5	-
No (reference)	96.2	42.9	51.8	29.9	10.2	46.4	-
Education (grade attended)	4.5	1.5	4.3	1.4	4.1	1.4	0.483
Household Income (dollars)	6.4	2.6	4.9	3.0	5.1	2.2	-
Gender ^a^	-	-	-	-	-	-	0.005
Male (reference)	102.6	45.8	52.2	30.2	9.2	41.8	-
Female	121.2	54.1	120.6	69.7	13.2	60.0	-
Age (years)	48.5	16.9	37.1	11.7	33.2	14.9	<0.001

^a^ Number (*n*) and percentage are reported.

**Table 2 healthcare-10-02444-t002:** Household income.

Income	*N*	%
Less than $10,000	26	6.2
$10,000 to under $15,000	25	6.0
$15,000 to under $20,000	33	7.9
$20,000 to under $25,000	22	5.3
$25,000 to under $35,000	40	9.5
$35,000 to under $50,000	49	11.7
$50,000 to under $75,000	47	11.2
$75,000 to under $100,000	38	9.1
$100,000 to under $150,000	36	8.6
$150,000 to under $200,000	11	2.6
$200,000 or more	13	3.1
Total	340	81.1

This table describes the income of participants. Income is a covariate in the final models and is treated as a continuous variable in the models. Just under one fifth of respondents (*n* = 79, 18.9%) did not know or declined to answer this question.

**Table 3 healthcare-10-02444-t003:** Negative Binomial Regression Coefficients to Predict Number of Media Exposure to Breastfeeding Promotion with Multiple Imputation (*n* = 419).

	*B*	*SE B*	*p*-Value
Model			
Constant	1.100	0.241	<0.001
African American	0.276	0.103	0.007
Other Race	0.032	0.214	0.881
Employed	0.149	0.101	0.139
Education	−0.039	0.036	0.277
Household Income	0.008	0.026	0.768
Female	0.390	0.100	<0.001
Age	−0.008	0.003	0.016
Likelihood Ratio Chi-square	41.627		<0.001

*B* = regression coefficient; *SE B* = standard error of the coefficient.

**Table 4 healthcare-10-02444-t004:** Ordinary Least Squares Regression Coefficients to Predict Prevalence of Breastfeeding with Multiple Imputation (*n* = 419).

	*B*	*SE B*	*p*-Value	*β*
Model				
African American	−1.010	0.199	<0.001	−0.311
Other Race	0.070	0.364	0.845	0.010
Employed	0.453	0.193	0.058	0.137
Education	0.104	0.062	0.083	0.094
Household Income	0.003	0.049	0.644	0.005
Female	0.104	0.173	0.407	0.032
Age	0.000	0.006	0.902	−0.005

*B* = unstandardized regression coefficient; *SE B* = standard error of the coefficient; *β* = standardized coefficient; *R*^2^ = 0.131 (coefficient of determination); Δ*R*^2^ = 0.116 (adjusted *R*^2^).

## Data Availability

Data available on request due to restrictions (e.g., privacy or ethical). The data are not publicly available due to data held in a private repository.

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
