# Peer review of "Racial Differences in Breastfeeding on the Mississippi Gulf Coast: Making Sense of a Promotion-Prevalence Paradox with Cross-Sectional Data"

_healthcare, 2022, doi:10.3390/healthcare10122444_

Round 1

Reviewer 1 Report

Thank you for entrusting me with this manuscript review. The topic of this research is very interesting, but there are some points that I think need to be clarified or revised in order to make this manuscript better.

1. The type of research (cross sectional) should be included in the title

2. Check again whether the writing is in accordance with the STROBE guidelines

3. This study wants to analyze racial differences, but the introduction of the basics why these racial differences want to be analyzed is lacking.

4. Continuing from point 3. The problem in the background is that African American women are assumed not to breastfeed. Why not just be specific stated in the title then another race as a control.

5. The results and discussion of the author are more focused on african american

That's my review, hope it helps

Author Response

We are grateful for the thoughtful comments that the reviewer has provided on the earlier draft of our manuscript. We have integrated extensive revisions, as is evident from the tracked changes in the revised manuscript. We have made every effort to follow the reviewer's recommendations. Here, we have provided reviewer comments followed by sections demarcated AUTHOR RESPONSE and italicized for point-by-point engagement with each review.

REVIEWER 1

Thank you for entrusting me with this manuscript review. The topic of this research is very interesting, but there are some points that I think need to be clarified or revised in order to make this manuscript better.

  1. The type of research (cross-sectional) should be included in the title

AUTHOR RESPONSE: The title has been revised to include “with cross-sectional data.”

  1. Check again whether the writing is in accordance with the STROBE guidelines

AUTHOR RESPONSE: STROBE Guidelines have been reviewed alongside the revised manuscript. Our manuscript is compliant with STROBE Guidelines.

  1. This study wants to analyze racial differences, but the introduction of the basics why these racial differences want to be analyzed is lacking.

AUTHOR RESPONSE: Every effort has been made to revise the introduction with additional emphasis on the need to analyze racial differences in breastfeeding. Specifically, we now more fully describe the magnitude and persistence of differences in breastfeeding among racial groups, with special attention to African American patterns. In addition, we more strongly justify our choice of the study setting, namely, the Mississippi Gulf Coast. Finally, we have returned to this important issue in the revised discussion by elaborating on the implications of our findings for African American women while also identifying fruitful paths for future research on breastfeeding for this focal population.

  1. Continuing from point 3. The problem in the background is that African American women are assumed not to breastfeed. Why not just be specific stated in the title then another race as a control.

AUTHOR RESPONSE: We have carefully revised our prose to avoid the mistaken assumption that African American women do not breastfeed. While rates of breastfeeding differ among racial-ethnic groups, we are now careful to review the nuances in breastfeeding disparities among racial-ethnic groups. More specifically, we now review facilitators that can bolster breastfeeding rates among African American women. We hope this more balanced approach avoids the assumption that African American women as a group do not breastfeed.

  1. The results and discussion of the author are more focused on african American

AUTHOR RESPONSE: The African American population in Mississippi is this study’s primary focus. We now further justify the need to focus on this population in the revised and expanded introduction. The revised manuscript discusses in more detail breastfeeding disparities related to race-ethnicity and, more specifically, rates among African American women.

That's my review, hope it helps

AUTHOR RESPONSE: Your comments were very helpful. Thank you!

Reviewer 2 Report

Bartkowski and colleagues have done an amazing work examining the African American and white racial disparities concerning (1) exposure to breastfeeding promotional information and (2) reported prevalence of breastfeeding in primary social networks. The manuscript is worthy of publication following revision based on the comments noted below:

Abstract:

It is not clear what the findings of the study are. The authors are suggested to add the results to the abstract as well.

Introduction:

Line 37: The authors indicate that the rates of African American women’s breastfeeding initiation and duration are lower. Provision of statistics regarding this would be useful. 

Line 41: Also, figures related to isolated cases of high BF rates would be useful, and any indication to the reason for the high rates as well.

Line 97: The research gaps and the rationale for this study could be further explored. 

Material and Methods:

The information on the setting and context of this study is useful.

Results:

The results are well-presented.

Discussion:

The discussion is extremely brief and could be significantly improved. It is not evident from the discussion what racial disparities exist, or the reasons for the differences or the reasons that could explain the low BF prevalence among African American women than their white peers. These need to be discussed in detail. 

Have the authors considered the role of maternal characteristics such as maternal age, education, socioeconomic status, and employment status, which have shown to be associated with breastfeeding duration. This could be integrated in the discussion, please refer to the papers below:

https://www.mdpi.com/1660-4601/17/15/5384/htm

https://www.ncbi.nlm.nih.gov/pmc/articles/PMC6202780/

While there are breastfeeding issues that have been identified among the general population, such as latching issues, were there any issues specific to the African American women that were identified in this study? These should be explored and discussed to better understand the racial differences. 

Line 318: For future directions, the need for a qualitative study could also be indicated. It is also essential for the authors to highlight the strengths of the study.

The authors are recommended to add a section on “implications for policy and practice” to highlight the implications of the findings of this study.

Author Response

We are grateful for the thoughtful comments that the reviewer has provided on the earlier draft of our manuscript. We have integrated extensive revisions, as is evident from the tracked changes in the revised manuscript. We have made every effort to follow the reviewer's recommendations. Here, we have provided reviewer comments followed by sections demarcated AUTHOR RESPONSE and italicized for point-by-point engagement with each review.

REVIEWER 2

Bartkowski and colleagues have done an amazing work examining the African American and white racial disparities concerning (1) exposure to breastfeeding promotional information and (2) reported prevalence of breastfeeding in primary social networks. The manuscript is worthy of publication following revision based on the comments noted below:

Abstract:

It is not clear what the findings of the study are. The authors are suggested to add the results to the abstract as well.

AUTHOR RESPONSE: The findings of this study are now featured as specific results conveyed within the abstract.

Introduction:

Line 37: The authors indicate that the rates of African American women’s breastfeeding initiation and duration are lower. Provision of statistics regarding this would be useful. 

AUTHOR RESPONSE: Our initial exclusion of background statistics for African American women’s breastfeeding initiation and duration from the introduction’s literature review was intended to convey information as concisely as possible. We now provide statistics from previous research that underscores the magnitude of breastfeeding disparities across racial-ethnic groups. Our own statistical analyses are now positioned as an extension of empirical results featured in previous studies.

Line 41: Also, figures related to isolated cases of high BF rates would be useful, and any indication to the reason for the high rates as well.

AUTHOR RESPONSE: In revising the literature review, we now have included research on facilitators of breastfeeding among African American women. This more balanced summary of previous research considers the contexts in which African American women’s breastfeeding rates may be elevated. We also discuss the specific contributors to these elevated rates when they are observed.

Line 97: The research gaps and the rationale for this study could be further explored. 

AUTHOR RESPONSE: To be as concise as possible, we were rather terse in identifying research gaps and specifying our study’s rationale. We are grateful for the opportunity to provide more information on each of these important fronts and have done so in the revised introduction.

Material and Methods:

The information on the setting and context of this study is useful.

AUTHOR RESPONSE: We are grateful for your comment about the usefulness of the materials and methods applied in this study.

Results:

The results are well-presented.

AUTHOR RESPONSE: We are grateful for your comments on this section of the manuscript.

Discussion:

The discussion is extremely brief and could be significantly improved. It is not evident from the discussion what racial disparities exist, or the reasons for the differences or the reasons that could explain the low BF prevalence among African American women than their white peers. These need to be discussed in detail. 

AUTHOR RESPONSE: We welcome the opportunity to expand our discussion and have done so through the addition of several paragraphs in the revised manuscript. We are now careful to identify what racial disparities exist while pinpointing impediments to breastfeeding among African American women. Our more in-depth treatment of these issues pays greater attention to structural inhibitors to breastfeeding among African American women. We are now also able to recommend possible solutions to these breastfeeding barriers.

Have the authors considered the role of maternal characteristics such as maternal age, education, socioeconomic status, and employment status, which have shown to be associated with breastfeeding duration. This could be integrated in the discussion, please refer to the papers below:

https://www.mdpi.com/1660-4601/17/15/5384/htm

https://www.ncbi.nlm.nih.gov/pmc/articles/PMC6202780/

AUTHOR RESPONSE: We are grateful for these references and have included them in the revised manuscript. Our study examines structural barriers to breastfeeding. Our statistical models account for maternal socioeconomic status and employment status, but we recognize that these issues could be treated in more detail through the revised discussion. Additional attention to maternal age and education has been included in the discussion, as suggested.

While there are breastfeeding issues that have been identified among the general population, such as latching issues, were there any issues specific to the African American women that were identified in this study? These should be explored and discussed to better understand the racial differences. 

AUTHOR RESPONSE: Our revised literature review now accounts for latching and related issues that can serve as impediments to breastfeeding. Where possible, we pay attention to racial-ethnic differences in these impediments. Regrettably, our study’s data do not feature indicators on these issues. However, we now suggest that these considerations be examined through future research.

Line 318: For future directions, the need for a qualitative study could also be indicated. It is also essential for the authors to highlight the strengths of the study.

AUTHOR RESPONSE: We enthusiastically agree that qualitative research could push the field significantly forward. We have expanded our discussion to indicate some lines of inquiry that might be fruitfully pursued through various qualitative methods. In addition, the strengths of our study are now more clearly specified in the expanded discussion section.

The authors are recommended to add a section on “implications for policy and practice” to highlight the implications of the findings of this study.

AUTHOR RESPONSE: We have now added a section on implications for policy and practice. We appreciate the opportunity to highlight practical applications of our research, especially those that could result in the adoption of more breastfeeding-friendly policies.